# Effects of Different Fermented Feeds on Production Performance, Cecal Microorganisms, and Intestinal Immunity of Laying Hens

**DOI:** 10.3390/ani11102799

**Published:** 2021-09-25

**Authors:** Lijuan Guo, Jing Lv, Yinglu Liu, Hui Ma, Bingxu Chen, Keyang Hao, Jia Feng, Yuna Min

**Affiliations:** College of Animal Science and Technology, Northwest A&F University, Yangling 712100, China; glj1430151801@163.com (L.G.); lvjing96@163.com (J.L.); liuyinglu1996@163.com (Y.L.); mahui1996@126.com (H.M.); chen3532946601@126.com (B.C.); goodkeyang@163.com (K.H.); fengjiacaas@163.com (J.F.)

**Keywords:** fermented feed, layer, production performance, cecal microbiomes

## Abstract

**Simple Summary:**

Fermented feed exerts beneficial effects on intestinal microorganisms, host health, and production performance. However, the effect of fermented feed on laying hens is uncertain due to the different types of inoculated probiotics, fermentation substrates, and fermentation technology. Hence, this experiment was conducted to investigate the effects of fermented feed with different compound strains on the performance and intestinal health of laying hens. Supplement fermented feed reduced the feed conversion ratio and promoted egg quality. Both dietary treatment (fermented feed A produced *Bacillus subtilis*, *Lactobacillus*, and *Yeast* and fermented feed B produced by *C. butyricum* and *L. salivarius*) influenced intestinal immunity and regulated cecal microbial structure. This may be because the metabolites of microorganisms in fermented feed and the reduced pH value inhibited the colonization of harmful bacteria, improved the intestinal morphology, and then had a positive impact on the production performance and albumen quality of laying hens.

**Abstract:**

This experiment was conducted to investigate the effects of different compound probiotics on the performance, cecal microflora, and intestinal immunity of laying hens. A total of 270 Jing Fen No.6 (22-week-old) were randomly divided into 3 groups: basal diet (CON); basal diet supplemented with 6% fermented feed A by *Bacillus*
*subtilis,*
*Lactobacillus*, and *Yeast* (FA); and with 6% fermented feed B by *C. butyricum* and *L. salivarius* (FB). Phytic acid, trypsin inhibitor, β-glucan concentrations, and pH value in fermented feed were lower than the CON group (*p* < 0.05). The feed conversion ratio (FCR) in the experimental groups was decreased, while albumen height and Haugh unit were increased, compared with the CON group (*p* < 0.05). Fermented feed could upregulate the expression of the signal pathway (TLR4/MyD88/NF-κB) to inhibit mRNA expression of pro-inflammatory cytokines (*p* < 0.05). Fermented feed promoted the level of *Romboutsia* (in the FA group) *Butyricicoccus* (in the FB group), and other beneficial bacteria, and reduced opportunistic pathogens, such as *Enterocooccus* (*p* < 0.05). Spearman’s correlations showed that the above bacteria were closely related to albumen height and intestinal immunity. In summary, fermented feed can decrease the feed conversion ratio, and improve the performance and intestinal immunity of laying hens, which may be related to the improvement of the cecal microflora structure.

## 1. Introduction

Corn and soybean meal (SBM) have been widely applied to animal feed because of their high nutritional value and organic matter digestibility. However, various anti-nutritional factors (ANFs) in feed, such as phytic acid and trypsin inhibitor (TI), decrease the digestibility and absorption of animals [1,2]. These ANFs may compromise the nutritional value, utilization, and digestibility of feed materials [3]; cause animal digestive and metabolic diseases; destroy the intestinal micro-ecological balance; and thereby have adverse effects on animal growth [4,5]. Fermentation is a feasible method by using microorganisms to decompose and transform the polysaccharides, ANFs, and other substances that are not easy to be digested and absorbed by livestock and poultry. As a consequence, fermented feed (FF) increased the average daily gain (ADG) of broilers and decreased the feed conversion ratio (FCR), which was beneficial to the growth performance of broilers, particularly in the later stage [6]. Moreover, it has been proved that FF could balance mucosal microflora [7] and promote immune function by activating the TLR-4-mediated MyD88-independent signaling pathway [8] in broilers.

Different fermentation strains have different reactions with substrate and produce different fermentation products. *Bacillus subtilis* (*B. subtilis*) has the ability to create an anaerobic environment in the intestinal tract, which was conducive to the growth and proliferation of lactic acid bacteria. Through competitive colonization, *B. subtilis* can eliminate and limit pathogenic bacteria in the intestinal tract [9]. Jin et al. [10] demonstrated FF produced by mixed probiotics, including *L. salivarius* and *B. subtilis*, was enriched with potential antimicrobial metabolites. *B. subtilis* eliminated 92.36% of glycinin and 88.44% of β-conglycinin from SBM [11]. The inoculated microorganisms remarkably change the production of FF [12,13,14,15,16]. *Clostridium butyricum* (*C. butyricum*) can consume dietary fiber and produce short-chain fatty acids (SCFAs), especially butyric acid, which can nourish intestinal goblet cells and reduce the number of intestinal pathogens [17,18]. Broilers supplemented *C. butyricum* had high average daily feed intake (ADFI) and ADG in the whole test period, and improved intramuscular fat content in the breast muscle at 42 days old [19]. Moreover, studies have shown that *C. butyricum* would upregulate the content of *Lactobacillus*, *Bifidobacterium*, and some other certain beneficial bacteria [20,21,22,23,24]. *Lactobacillus salivarius* (*L. salivarius*) promoted the growth performance of hens by protecting the gut from pathogens and improving the extraction of nutrients and energy [25]. Previous reports have confirmed that both *C. butyricum* and *L. salivarius* have strong adhesion to intestinal epithelium and produce bacteriocin, thus promoting the rapid growth of beneficial bacteria and reducing the colonization of pathogens [26,27,28]. The combined use of probiotics produced a better effect [29]; however, the effect of the combined use on laying hens remains to be determined in further studies.

Our team previous study has shown that adding 6% FF to the basal diet had the best effects on the production performance and intestinal barrier function of laying hens. However, due to the different fermentation substrates, inoculating strains, and product characteristics, the quality of fermented feed and its effects on the performance of poultry are different. In addition, most of the current experiments have focused on broilers and there are few studies of laying hens. Therefore, the most commonly used feed materials: corn, bran, and SBM, were selected in this study to investigate the effects of two compound probiotics FF on the gut health of laying hens.

## 2. Materials and Methods

All experimental procedures were conducted in accordance with the institutional animal care and committee (IACUC) guidelines and were approved by the Animal Care and Use Committee of Northwest A&F University, Yangling, China (NWAFU-314020038).

### 2.1. Preparation of Fermented Feed

Compound bacteria I (*B. subtilis* 2 × 10^9^ CFU/g, *Lactobacillus* 3 × 10^9^ CFU/g, *Yeast* 5 × 10^8^ CFU/g) were provided by Shandong Baide Co., Ltd. (Weifang, China).

Compound bacteria II included *C. butyricum* 2 × 10^8^ CFU/g (provided by Guangdong Dazenong Biotechnology Co., Ltd. Guangdong, China) and *L. salivarius* 1 × 10^8^ CFU/g (isolated from lab). 

Mixed diet was composed of 60% corn, 20% wheat bran, and 20% soybean meal. 

Firstly, the frozen-dried powder of probiotics powder was dissolved in warm water and stirred evenly to obtain the diluted mixed bacteria solution. Secondly, sterile water was supplemented into mixed diet, which poured dilute bacteria for a 30% moisture content. Thirdly, the mixed diet was put into a sealed bag, and the air was exhausted. Anaerobic fermentation was allowed to occur at 37 °C for 5 days. Finally, 6% FF was added to the basal diet to obtain the experimental diet. The FF was prepared once a week and adjusted according to the feed intake of layers. Feed samples were analyzed to establish the contents of routine nutrients and pH value according to AOAC International guidelines. Phytic acid, TI, and β-glucan in feed were determined using a commercial kit (Jianglai Bio Company, Shanghai, China). 

### 2.2. Experimental Design

A total of 270 Jing Fen No.6 laying hens (22-weeks-old) with a similar body weight and good health were randomly divided into 3 groups with 6 replicates in each group and 15 hens in each replicate. Three dietary treatments were as follows: basal diet (CON), basal diet supplemented with 6% compound bacteria I FF (FA), and basal diet supplemented with 6% compound bacteria II FF (FB). The experimental period was 8 weeks. According to the feeding standard of NRC (1994) and the nutritional level recommended by Hailan Company, the corn-SBM-based diet was designed (shown in Table 1).

### 2.3. Animal Management

This experiment was carried out in Demonstration Farm of Non-resistance Breeding of Chunmanyuan Layers in Tongchuan District of Shaanxi Province. Experimental layers were housed in cages with 16 h of light. At the beginning of the trial, the laying rate of chickens was 87.62 ± 2.67%. During the experiment, birds were given ad libitum access to the diet and water. The temperature in the house was maintained at 15~22 °C and humidity at 30~50%. At the end of the experiment, one healthy bird in each replicate was randomly selected and euthanized by exsanguination. The intestinal segments (duodenum, jejunum, and ileum) and cecal contents were collected and immediately frozen in liquid nitrogen, and then stored at −80 °C to be measured.

### 2.4. Production Performance and Egg Quality

The eggs collected at 17:00 every day were weighed and counted. Remaining feed was weighed in the morning every weekend, and the average daily feed intake (ADFI), laying rate, and feed conversion ratio (FCR) were calculated. Two eggs were randomly selected from each replicate at the end of the experiment. Albumen height (AH), yolk color, and Haugh unit (HU) were determined by an egg analyzer (EMT-5200, Robotmation, Japan). Eggshell strength and eggshell thickness were measured using an eggshell strength analyzer (EFG-0503, Robotmation, Japan) and eggshell thickness analyzer (EFG-1061, Robotmation, Japan), respectively. The vertical and horizontal diameters of eggs were measured by a Vernier caliper, and the egg shape index was calculated.

### 2.5. Intestinal Morphology

At the end of the experiment, about 2-3 cm as taken from the duodenum, jejunum, and middle ileum; the chyme was rinsed with a syringe containing 0.9% normal saline; and then was fixed in a 10 mL centrifuge tube with 4% paraformaldehyde solution. Then, 5-mm-thick intestinal tissue was selected to make sections and stored at room temperature. The fixed segments were sectioned and observed under the light microscope, and the villus height (VH) and crypt depth (CD) of each segment were measured.

### 2.6. RNA Extraction and Polymerase Chain Reaction (PCR) Amplification

Total RNA was extracted from each jejunal mucosa sample using a CWBIO^®^ Ultrapure RNA Kit according to the manufacturer’s instructions. The total RNA concentration was detected by a microplate reader (Bio-Tek, Winooski, VT, USA), and 1% of agarose gel electrophoresis was used to determine RNA purity. RNA samples were diluted to 1 ng/μL and subsequently reverse transcribed into cDNA. Diluted cDNA was used to measure the expression levels of ZO-1, Occludin, and Mucin-2, TNF-α, IL-8, TLR 4, and related signal proteins MyD88 and NF-κB. Primers used for quantitative real-time PCR are shown in Table 2.

### 2.7. 16S rDNA Gene Sequencing and Bioinformatics Analysis

Genomic DNA of the cecal content was extracted by the CTAB method. Agarose gel electrophoresis was used to detect the purity and concentration of DNA. An appropriate sample of DNA was applied to the centrifuge tube and diluted with sterile water to 1 ng/μL. Diluted genomic DNA as template and phusion ^®^ High fidelity PCR master mix with GC buffer (New England Biolabs) was used for PCR amplification. PCR products were detected by electrophoresis with 2% agarose gel. A Truseq ^®^DNA PCR-Free Sample Preparation Kit provided by Illumina was used to construct the gene library. After Qubit and Q-PCR quantitation, the qualified library was sequenced by novaseq6000. The samples were clustered into operational taxonomic units (OTUs) using Uparse software (Uparse v7.0.1001, http://www.drive5.com/uparse/, accessed on 29 August 2020), and the representative principle of OTUs was selected according to the algorithm principle. The Chao1, Shannon, and ACE index were calculated with Qiime software (version 1.9.1) to analyze the microbial alpha diversity. Using the UniFrac phylogenetic distance, the bacterial diversity was assessed to detect the structural changes of microbial communities among the treatment groups, and principal coordinate analysis (PCoA) was used for visualization.

### 2.8. Statistical Analysis

In order to further search for the species of different groups under different classification levels (phylum, family, genus), and to perform the *T*-test among the groups, species with significant differences were found. The Spearman correlation coefficient was used to evaluate the changes in the production performance, intestinal immunity, and microbiota. Data of laying hens were analyzed by one-way ANOVA and Duncan’s multiple range test using SPSS 26.0 (SPSS Inc., Chicago, IL, USA). Results are presented as means with standard error of the mean (SEM). *p*-value of less than 0.05 was considered statistically significant.

## 3. Results

### 3.1. Fermented Feed Characteristics

As shown in Table 3, the content of crude protein (CP) in FA and FB was 15.08% and 13.09% higher than CON, respectively (*p* < 0.05). Crude fiber (CF) and pH value also decreased significantly in FF (*p* < 0.05). Moreover, the concentration of TI in FA and FB decreased by 2.48% and 11.66%, respectively, during fermentation (*p* < 0.05). Furthermore, the content of β-glucan in FA and FB was 202.38 and 183.33 pg/mL respectively, which was significantly different from 207.52 pg/mL in the CON group (*p* < 0.05). After fermentation, the number of viable bacteria in the FA group was 1.74 × 10^7^ CFU/g for *B. subtilis*, 2.33 × 10^5^ CFU/g for *Yeast*, and 1.62 × 10^7^ CFU/g for *Lactobacillus*, respectively, while that in the FB group was 1.32 × 10^7^ CFU/g for *C. butyricum* and 1.89 × 10^7^ CFU/g for *L. salivarius*, respectively.

### 3.2. Laying Performance and Egg Quality

The laying performance and egg quality are summarized in Table 4. Although no distinct difference was found for the ADFI, laying rate, yolk color, eggshell thickness, and strength, FF dramatically reduced the FCR (*p* < 0.05), compared with that in the CON group. Besides, dietary FF treatment increased the laying rate by 3%. In terms of egg quality, adding 6% of FF presented an upward trend in albumen height and Haugh unit (*p* < 0.05).

### 3.3. Effects of FF on the Intestinal Immune-Related Genes of Layers

The results shown in Table 5 reveal that no noteworthy effect was observed on the concentration of ZO-1 (*p >* 0.05). However, the level of Mucin-2 in the FB group was extremely upregulated, compared with the CON group and FA (*p* < 0.05). Furthermore, compound bacteria FF made the concentration of Occludin higher than the CON group (*p* < 0.05), and no notable differences were found between FA and FB *(p >* 0.05). The data indicated that FF can decrease the content of pro-inflammatory genes IL-8 and TNF-α (*p* < 0.05). Supplementation of 6% FF significantly upregulated the content of TLR4 and MyD88 (*p* < 0.05), while striking promoted the NF-κB gene expression level (*p* < 0.05). Furthermore, the effects of FB were superior to FA.

### 3.4. Intestinal Morphology

In order to quantitatively evaluate intestinal morphology, the commonly used indexes for measuring the intestinal histology of hens were determined, as shown in Table 6, including villus height (VH), crypt depth (CD), and their ratio (VH/CD). From the table, FA and FB significantly improved the integrity of intestinal mucosa. Compared with the control group, the VH of the duodenum and the VH/CD ratio significantly increased (*p* < 0.05). Besides, probiotics treatment promoted the VH of the jejunum, while FF greatly decreased the crypt depth so that made the VH/CD increase significantly (*p* < 0.05). 

### 3.5. Effects of FF on Cecal Microbial Composition

The community of gut diversity showed an upward trend (*p* = 0.068) in contrast with that in the control group. The increase of the Chao1 index and ACE index indicated the gut community richness significantly improved in the FB group (*p* < 0.05), compared with that of the FA and CON group. Additionally, as shown in Table 7, there were no distinct differences between the FA and CON group.

In order to further analyze the differences of the cecal microbial community structure between the control group and the treatment group, the PCoA method was used to analyze the cecal microbial community structure according to weight UniFrac. As presented in Figure 1, FB is mainly concentrated in the first quadrant, which was significantly different from FA and CON. From what has been discussed above, compound probiotics II FF could significantly improve the species and evenness of cecal microorganisms in layers. 

The results of sequencing revealed that the dominant bacteria in the layers’ caecum of the three groups were *Bacteroidetes*, *Firmicutes*, *Fusobacteria, Proteobacteria*, and *Euryarchaeota* (Table 8). Compared with the CON group, the RA of *Bacteroidetes* was significantly increased in both experimental groups (*p* < 0.05), and the RA of *Proteobacteria* tended to decrease in the experimental group (*p* = 0.07).

At the genus level (Figure 2a,b), fermented feed significantly influenced the intestinal microflora of laying hens. The RA of *Romboutsia* (in the FA group), *Butyricicoccus* (in the FB group), and other beneficial bacteria for intestinal health was significantly higher than that in the CON group (*p* < 0.05). The relative abundance of opportunistic pathogens, such as, *Alistipes**, Enterococcus,* and *A**lloprevotella*, decreased significantly in contrast with the CON group (*p* < 0.05).

### 3.6. Correlation of Laying Performance, Gut-Associated Gene Expression, and Gut Bacteria

Spearman correlation analysis was applied to investigate the relationship among laying performance, expression of gut-associated genes, and gut bacteria (Figure 3). The abundance of *Bacteroides* was positively corrected with albumen height and strongly negatively corrected with TNF-α. As to the gut-associated gene expression, *Alistipes* was negatively correlated with Mucin-2, TLR 4, and MyD88, and positively correlated with TNF-α and IL-8 (*p* < 0.05).

## 4. Discussion

The improvement of the chemical composition and nutritional value of poultry feed by fermentation has been widely reported. This feed processing technology aids the digestion and absorption of nutrients by the host and thereby improves animal growth performance [30]. The crude protein content, pH value, and living bacteria number are important indexes to evaluate the nutritional value of FF [31]. Our results showed that fermentation with probiotics could significantly increase the CP content in the substrate. This may be because the loss of dry matter, especially carbohydrates, in fermented feed increases the relative concentration of other nutrients. The increase of CP may also be due to the hydrolysis of macromolecular proteins, especially antigen proteins [32]. At the same time, a large amount of probiotics in fermented feed could produce positive effects on the host, including inhibiting the proliferation of pathogenic microorganisms [33] and improving the intestinal barrier function [34]. A previous study demonstrated that microbial fermentation of SBM efficiently eliminates ANFs and enhances the nutrient value [35,36], which was consistent with our results. It was observed that the contents of crude fiber, phytic acid, and β-glucan were reduced following fermentation with compound probiotics. This may be due to the production of related enzymes, such as cellulase and phytase, which degraded these anti-nutrient substrates. Therefore, the elevated CP content, lower pH value, and reduced anti-nutrient factors in response to fermentation would be beneficial to the gut health and production performance of laying hens.

In pig production, results showed that FF can improve ADFI, FCR, and production performance [11,37,38,39]. However, to date, the results in layers and broilers are limited and inconsistent. The data obtained in the present study demonstrated that adding compound probiotics to fermented feed significantly increased albumen height and Haugh unit and decreased FCR of layers, compared with the control group, but did not influence ADFI and other relevant indicators during the test stage, similar to Li [40]. Adding fermented SBM produced by *B. subtilis* could significantly improve the daily gain of broilers [41]. However, it was reported that probiotics had little effect on the growth performance of broilers [42]. This inconsistency may be related to the species of probiotics, preparation technology, dosage, feed composition, and bird age and health status. In addition, it has been reported that the combination of probiotics can exhibit synergistic effects [43]. The improvement of the growth performance and feed efficiency of layers by adding compound probiotics was considered to be caused by the cumulative effect of probiotics, such as *C. butyricum*, which can promote the growth of beneficial lactic acid-producing bacteria [44].

Albumen height (AH) and Haugh unit (HU) are important indexes of egg quality. Haugh unit is mainly determined by the content of thick protein [45], and the value represents the viscosity of egg protein: the higher the protein viscosity, the longer the egg stays fresh. AH and HU, as a measure of egg quality, may be related to protein synthesis and water transfer in yolk [46]. Feed with 1 × 10^8^ CFU/kg *B. subtilis* significantly increased AH, HU, and eggshell thickness [47], which was consistent with our results. The changes of AH and HU were influenced by the electrolyte balance of birds, suggesting that the increase of AH was related to the increase of the divalent cation concentration [48]. These may be attributed to the addition of *C. butyricum*, *L. salivarius*, and *Lactobacillus* to create a favorable microbial intestinal environment in birds, which decreased the lumen pH and enhanced the solubility and absorption of cations [49]. Probiotics reduced the pH value in the gut, which helps dissolve more Ca and P, and has a positive impact on Ca deposition into the eggshell. In addition, the increased CP content in fermented feed may be another factor responsible for the elevation of the albumen height and Haugh unit.

Gramicidin produced by *B. subtilis* and the stimulation of the maturation and differentiation of dendritic cells of *L. salivarius* stimulated immune regulation in a non-inflammatory way [50], maintained metabolic homeostasis, and ultimately regulated host performance. The jejunal-associated immune gene expression reflected the physiological and immune status of laying hens. Cytokines were divided into pro-inflammatory cytokines and anti-inflammatory cytokines. In this study, the expression of MyD88, NF-κB, and TLR-4 was significantly upregulated and the content of IL-8 and TNF-α were dramatically downregulated in the FF groups, in contrast with those in the CON group, suggesting that FF can regulate the immune function of laying hens. It was reported that the expression of pro-inflammatory genes was downregulated in response to fermented diets [51,52]. Similar results were found in this study. Tight junctions (TJS) are one of the ways in which cells connect to the endothelium, which plays a key role in the integrity of the barrier and the permeability of the endothelium [53]. TJS are composed of Occludin, Mucin-2, and ZO-1, which maintain the stability of intercellular junctions [54]. Our results showed that the expression of Occludin and Mucin-2 was significantly increased by FF. Similar to Al-Sadi et al. [55], they reported that most proinflammatory factors, such as IL-1β and TNF-α, can lead to destruction of the epithelial tight junction barrier. Pan et al. [56] reported that downregulation of inflammatory gene expression (IL-1, IL-6, IL-8, and TNF-α) was negatively correlated with the expression genes (Occludin, ZO-1, and Mucin-2) involved in tight junction protein biosynthesis. The TLR signaling pathway can be divided into two types: MyD88-dependent pathway and MyD88-independent pathway [57]. In the MyD88-independent signaling pathway, TRIF can only be recruited by TLR-3 and TLR-4 to activate the NF-κB signaling pathway [58]. The concentration of NF-κB, MyD88, and TLR-4 has a striking upward trend, indicating high expression of the classical pathway (NF-κB/MyD88/TLR-4) inhibited the expression of proinflammatory factors, so as to promote expression of tight junction protein and maintain intestinal health.

Previous evidence suggested that there is a close relationship between the growth performance and intestinal flora of different species of animals Erratum in [37] and [59]. VH, CD, and VH/CD are crucial in the intestinal health of animals, which are directly related to the absorption capacity of intestinal mucosa. Studies have reported that morphological changes of the small intestine, such as increasing VH and decreasing CD, enhanced gut digestion and absorption [60,61], which was also found in the current study. Our results showed that FF increased VH of each intestinal segment, decreased jejunal CD, and increased the VH/CD ratio. The main reason might be that the combination of probiotics reduced the relative abundance of *Enterococcus* and other pathogens at the genus level, which may injure intestinal mucosal epithelial cells (IECs). Another reason may be that butyric acid, the main metabolite of *C. butyricum*, can nourish intestinal goblet cells and repair and protect IECs [17] in the FB group. 

The composition of intestinal microorganisms could be modulated by dietary composition [62]. The mechanism of probiotics for improving the production performance and egg quality of hens may include the change of intestinal microbiome, partly by inhibiting the proliferation of pathogens and promoting the growth of non-pathogenic facultative anaerobes and Gram-positive bacteria producing lactic acid [63], and promoting the digestion and utilization of nutrients. Intestinal flora assumed enormous importance in nutrition, physiology, and mucosal morphology [64]. In the present study, ACE and Chao1 richness estimators were significantly increased in FB layers, indicating higher species richness in the cecal digestive tract. Intestinal microorganisms with a higher abundance and quantity are more favorable for animals to cope with different environmental disturbances [65]. Beta diversity was used to further estimate the changes in cecal microbiota with FF addition. The average distance between the test groups showed that different treatments had significant differences in the intestinal bacterial community of laying hens. This evidence showed that after adding *C. butyricum* and *L. salivarius* to the fermented feed, the intestinal microorganisms of laying hens gradually reached a stable state and dynamic equilibrium. However, there was no significant distinction between the FA and CON groups, which need to be investigated in a future study, and further analysis was conducted on alterations of the microbiota composition following FF addition.

At the phylum level, dominant bacteria in the caecum of all groups included *Bacteroidetes, Firmicutes, Fusobacteria, Proteobacteria*, and *Euryarchaeota*. In addition, feeding FF significantly increased RA of *Bacteroidetes,* and decreased RA of *Proteobacteria* at the phylum level, which was similar to that of Li et al. [40]. Several gut bacteria can utilize complex polysaccharides as a source of carbon and energy, and in this process, end products released by these bacteria also provided nutrition and other characteristics that were beneficial to the host [66]. Therefore, we speculated that the presence of a large number of *Bacteroides* in layers fed with FF may contribute to the adaptation of these components, so as to improve the digestibility of nutrients and improve the production performance. In this study, the RA of *Proteobacteria* in the two experimental groups was much lower than that in the CON group, which was consistent with previous studies [67]. Studies have shown that undigested dietary protein in the gut promotes reproduction of some harmful bacteria [68].Therefore, we conjectured that the low abundance of *Proteobacteria* might be related to the increased CP digestibility of fermented feed. Meanwhile, *Lactobacillus* in FF reduced the intestinal pH value by producing organic acids, and prevented the colonization of intestinal pathogens through competitive rejection, antagonistic activity, and bacteriocin production, which also helps to reduce the abundance of *Proteobacteria* [64]. 

Results of the *T*-test showed that cecal microorganisms could be modified in a different manner by these two kinds of FF at the genus level. The RA of *Romboutsia* was increased in the FA group, which was reported to be involved in carbohydrate utilization, single amino acid fermentation, anaerobic respiration, and metabolic end products. The RA of *Alistipes* was decreased in the FA group, a bacterial genus closely related to intestinal diseases. It has been isolated from appendicular and brain abscesses, highlighting their potential opportunistic role [69]. Supplementation of *C. butyricum* and *L. salivarius* resulted in an increasing trend of some beneficial bacteria, such as *Butyricicoccus,* which can balance the structure of mucosal flora, produce butyric acid with anti-inflammatory properties, and strengthen the intestinal epithelial barrier, thus favoring intestinal health [70]. The reduced RA of *Alloprevotella* and *Gallibactergiella* and other conditional pathogens would alleviate local or systemic inflammation, contributing to the improved growth performance [71]. 

Correlation analysis revealed the relationship between production performance, gut immune function, and intestinal microorganisms, and clarified the potential roles of intestinal microorganism structure in the production performance and gut immune functions of laying hens. It was worth noting that an increase in *Bacteroides* abundance was accompanied by the improvements of albumen height, which was also discovered in previous studies [72]. *Bacteroides* is widely regarded as one of the beneficial genera, which is an effective degraders of nondigestible carbohydrates and SCFA producers, and plays an important role in the maturation of the immune system [73]. Besides, some bacteria, such as *Roseburia, Butyricicoccus, Intestinimonas*, and *Faecalibactium*, were reported to participate in regulation of the immune response and their close relationship was also confirmed in our study. In addition, a decreased abundance of *Alistipes* was accompanied by an enhancement of tight junction protein. Therefore, the intestinal microbiota composition could be modified by FF addition, while the richness of microbiota was only enriched in the FB group, which may contribute to the improved gut health and production performance of laying hens.

## 5. Conclusions

In conclusion, fermented feed produced by compound probiotics (FA, *B. subtilis, Lactobacillus* and *Yeast*; FB, *C. butyricum* and *L. salivarius*) improved the intestinal morphology, epithelial barrier functions, and immune status, thus favoring feed efficiency and albumen quality of laying hens. The greater performance and improved gut health could be in part explained by the enhancement of bacterial richness in the FB group and minor microbiota shifts characterized by the enrichment of *Bacteroidetes* in both the FA and FB groups. These findings may provide insights into the regulatory roles of fermented feed in laying hens.

## Figures and Tables

**Figure 1 animals-11-02799-f001:**
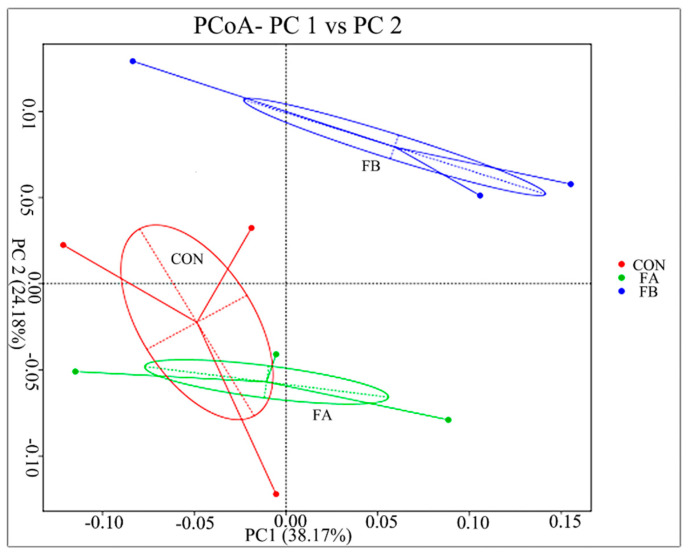
Bacterial diversity analysis of cecal microflora of laying hens fed different fermented diets (CON, red; FA, green; FB, blue) using principal coordinate analysis (PCoA) based on weighted UniFrac distances.

**Figure 2 animals-11-02799-f002:**
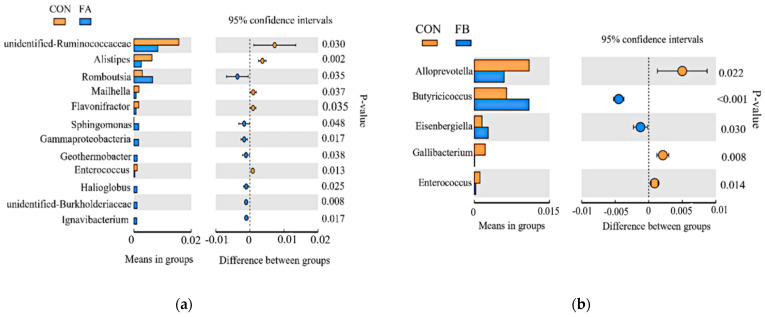
The relative abundance of bacterial genera in the cecal microbiota of layers in different groups ^1^. (**a**) *T*-test of cecal contents at the genus level between CON and FA; (**b**) T-test of cecal contents at the genus level between CON and FB. CON, control group; FA, basal diet with 6% FF produced by *B. subtilis*, *Lactobacillus*, *Yeast*; FB, basal diet with 6% FF produced by *C. butyricum*, *L. salivarius*.

**Figure 3 animals-11-02799-f003:**
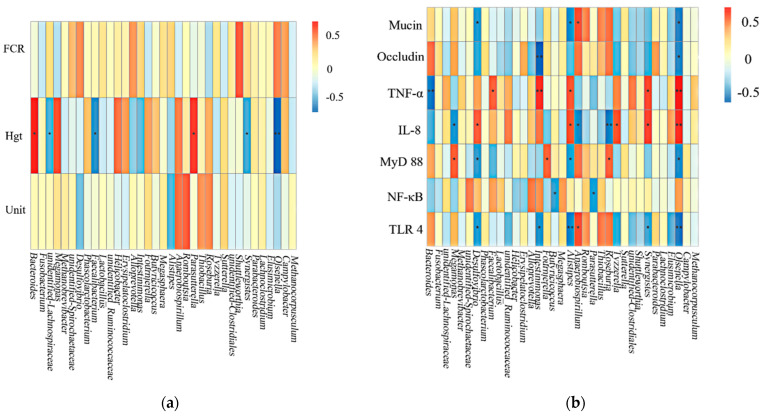
Correlation of laying performance ^1^, gut-associated gene expression ^2^, and gut bacteria. * *p* < 0.05, ** *p* < 0.01. Red represents a positive correlation, and blue represents a negative correlation. (**a**) Effects on laying performance. FCR, feed conversion ratio; Hgt, Albumen height; Unit, Haugh unit. (**b**) Effects on gut-associated gene. ^2^ Mucin, Mucin-2; TNF-α, Tumor necrosis factor –α; IL-8, Inflammatory mediators interleukin-8; MyD88, Myeloid differentiation primary response; TLR-4, Toll-like receptor 4.

**Table 1 animals-11-02799-t001:** Composition and nutrient levels of the basic diet (air-dry basis, %).

Ingredients	%	Nutrients ^1^	%
corn	60.80	ME (MJ/kg)	12.09
SBM	22.20	Crude protein	14.13
Stone powder	7.50	Calcium	4.99
Wheat bran	4.00	Total phosphorus	0.42
Soybean oil	0.50	Available phosphorus	0.32
Premix ^2^	5.00	Lys	0.94
Total	100.00	Met	0.44

^1^ Crude protein was measured, and the rest was calculated. ^2^ The premix provided the following per kg of diets: VA 10,000 IU, VD31, 800 IU, VE 10 IU, VK 10 mg, VB 125 ug, thiamine l mg, riboflavin 4.5 mg, calcium pantothenate 50 mg, niacin 24.5 mg, pyridoxine 5 mg, biotin 1 mg, folic acid 1 mg, choline 500 mg, Mn 60 mg, I 0.4 mg, Fe 80 mg, Cu 8 mg, Se 0.3 mg.

**Table 2 animals-11-02799-t002:** Primers used for quantitative real-time PCR in this study.

Gene ^1^	Primers Sequence ^2^ (5′–3′)
β-actin	F: ACACCCACACCCCTGTGATGAA
	R: TGCTGCTGACACCTTCACCATTC
TNF-α	F: AGTGCTGTTCTATGACCGCC
	R: CGCTCCTGACTCATAGCAGA
IL-8	F: CTGCGGTGCCAGTGCATTAG
	R: GCACACCTCTCTTCCATCC
NF-κB	F:TCAATGGCTACACAGGACCA
	R:CACTGTCACCTGGAAGCAGA
TNF-α	F: AGTGCTGTTCTATGACCGCC
	R: CGCTCCTGACTCATAGCAGA
TLR4	F:GCCATTGCTGCCAACATCATCC
	R:ATGCCAGAGCGGCTACTCAGAA
MYD88	F:CGTCGCATGGTGGTGGTTGTT
	R:TCGCTTCTGTTGGACACCTGGA
ZO-1	F:TATAGAAGATCGTGCCGCCTCC
	R:GAGGTCTGCCATCGTAGCTC
Occludin	F:ACAGCCCTCAATACCAGGATGTG
	R:ACCATGCGCTTGATGTGGAA
Mucin-2	F: TTCATGATGCCTGCTCTTGTG
	R: CCTGAGCCTTGGTACATTCTTGT

^1^ ZO-1, zonula occludens; IL-8, Inflammatory mediators interleukin-8; TNF-α, Tumor necrosis factor –α; MyD88, Myeloid differentiation primary response; TLR-4, Toll-like receptor 4.; ^2^ F = forward primer; R = reverse primer.

**Table 3 animals-11-02799-t003:** The nutrient composition of feed before and after fermentation.

Items ^1^	Groups ^2^			SEM	*p*-Value
	CON	FA	FB		
Crude protein, %	14.06 ^b^	15.90 ^a^	16.18 ^a^	0.35	<0.01
Ash, %	2.80 ^a^	2.36 ^b^	2.23 ^c^	0.09	<0.01
Ether extract, %	1.65	1.59	1.67	0.08	0.93
Crude fiber, %	4.45 ^a^	3.73 ^b^	3.47 ^b^	0.16	<0.01
β-Glucan, pg/mL	207.52 ^a^	202.38 ^b^	183.33 ^c^	3.24	<0.01
TI, μg/g	219.24 ^a^	214.72 ^b^	212.32 ^b^	1.05	<0.01
phytic acid, %	0.47 ^a^	0.44 ^b^	0.42 ^b^	0.01	0.04
pH	6.27 ^a^	4.94 ^b^	4.86 ^b^	0.23	<0.01
Viable count, CFU/g		*B.* 1.47 × 10^7^	*C.* 1.32 × 10^7^		
		*Y.* 2.33 × 10^5^			
		*LC.* 1.62 × 10^7^	*LS.* 1.89 × 10^7^		

^a,b,c^ Means within each row with different superscripts are statistically significantly different (*p* < 0.05). ^1^ TI, Trypsin inhibitor; *B.*, *B. subtilis*; *Y.*, *Yeast; LC.*, *Lactobacillus*; *C.*, *C. butyricum*; *LS*., *L. salivarius*; ^2^ CON, control group; FA, basal diet with 6% FF produced by *B. subtilis*, *Lactobacillus*, *Yeast*; FB, basal diet with 6% FF produced by *C. butyricum, L*. *salivarius*.

**Table 4 animals-11-02799-t004:** Effects of 6% FF on the performance and egg quality of laying hens.

Items ^1^	Groups ^2^			SEM	*p*-Value
	CON	FA	FB		
Performance					
ADFI, g	118.31	120.14	119.32	0.88	0.66
Average Egg weight, g	56.74	57.39	57.13	0.22	0.52
FCR	2.19 ^a^	2.02 ^b^	1.99 ^b^	0.03	0.03
laying rate, %	88.83	91.83	91.83	0.01	0.20
Egg quality					
Egg shaped index	0.79	0.76	0.80	0.01	0.09
Eggshell thickness, mm^2^	0.36	0.38	0.38	0.01	0.55
Eggshell strength, kg/cm^2^	37.86	39.04	38.73	0.45	0.61
Albumen height, mm	9.23 ^b^	9.60 ^a^	9.67 ^a^	0.09	0.01
Yolk color	6.43	6.75	7.25	0.19	0.20
Haugh unit	96.00 ^b^	97.70 ^ab^	98.91 ^a^	0.53	0.04

^a,b^ Means within each row with different superscripts are statistically significantly different (*p* < 0.05). ^1^ ADFI, average daily feed intake; FCR, feed conversion ratio, FCR= feed intake/egg mass; ^2^ CON, control group; FA, basal diet with 6% FF produced by *B. subtilis*, *Lactobacillus*, *Yeast*; FB, basal diet with 6% FF produced by *C. butyricum*, *L. salivarius*.

**Table 5 animals-11-02799-t005:** Effects of fermented feed on the mRNA levels of genes in the jejunum of laying hens.

Items ^1^	Groups ^2^			SEM	*p*-Value
	CON	FA	FB		
Tight junctions					
Mucin-2	1.01 ^b^	1.32 ^b^	1.76 ^a^	0.12	<0.01
Occludin	1.00 ^b^	2.34 ^a^	2.88 ^a^	0.29	<0.01
ZO-1	1.08	1.04	1.19	0.03	0.142
Immune-related genes					
IL-8	1.00 ^a^	0.39 ^b^	0.19 ^c^	0.12	<0.01
TNF-α	1.00 ^a^	0.52 ^b^	0.48 ^b^	0.09	<0.01
MyD88	1.02 ^c^	2.03 ^b^	2.66 ^a^	0.24	<0.01
NF-κB	1.05 ^b^	1.52 ^ab^	2.13 ^a^	0.19	0.046
TLR-4	1.01 ^c^	1.35 ^b^	1.76 ^a^	0.12	<0.01

^a,b,c^ Means within each row with different superscripts are statistically significantly different (*p* < 0.05). ^1^ ZO-1, zonula occludens; IL-8, Inflammatory mediators interleukin-8; TNF-α, Tumor necrosis factor –α; MyD88, Myeloid differentiation primary response; TLR-4, Toll-like receptor 4; ^2^ CON, control group; FA, basal diet with 6% FF produced by *B. subtilis*, *Lactobacillus*, *Yeast*; FB, basal diet with 6% FF produced by *C. butyricum*, *L. salivarius*.

**Table 6 animals-11-02799-t006:** Effect of fermented feed on the intestinal morphology of laying hens.

Items ^1^	Groups ^2^			SEM	*p*-Value
	CON	FA	FB		
Duodenum					
VH, μm	989.13 ^b^	1007.79 ^a^	1014.33 ^a^	4.18	0.03
CD, μm	151.17	150.76	150.93	0.43	0.94
VH/CD	6.54 ^b^	6.69 ^a^	6.72 ^a^	0.02	<0.01
Jejunum					
VH, μm	971.76 ^b^	1009.31 ^a^	1023.85 ^a^	6.83	<0.01
CD, μm	155.91 ^a^	152.01 ^b^	152.19 ^b^	0.62	<0.01
VH/CD	6.24 ^b^	6.64 ^a^	6.73 ^a^	0.07	<0.01
Ileum					
VH, μm	995.47	1008.23	1011.66	3.62	0.13
CD, μm	153.33	152.90	153.93	0.32	0.49
VH/CD	6.49 ^b^	6.59 ^a^	6.57 ^a^	0.02	0.03

^a,b^ Means within each row with different superscripts are statistically significantly different (*p* < 0.05). ^1^ VH, villus height; CD, crypt depth; VH/CD, villus height/crypt depth; ^2^ CON, control group; FA, basal diet with 6% FF produced by *B. subtilis*, *Lactobacillus*, *Yeast*; FB, basal diet with 6% FF produced by *C. butyricum*, *L. salivarius*.

**Table 7 animals-11-02799-t007:** Effect of fermented mixture on the cecal microbiota α-diversity of the laying hens.

Items ^1^	Groups ^2^			SEM	*p*-Value
	CON	FA	FB		
Chao1	610.67 ^b^	626.89 ^b^	964.58 ^a^	61.01	<0.01
ACE	591.04 ^b^	614.91 ^b^	938.91 ^a^	58.56	<0.01
Shannon	6.46	6.39	6.66	0.12	0.68

^a,b^ Means within each row with different superscripts are statistically significantly different (*p* < 0.05). ^1^ Chao 1 and ACE index were used to calculate the community richness; Shannon index was used to indicate the community diversity; ^2^ CON, control group; FA, basal diet with 6% FF produced by *B. subtilis*, *Lactobacillus*, *Yeast*; FB, basal diet with 6% FF produced by *C. butyricum*, *L*. *salivarius*.

**Table 8 animals-11-02799-t008:** Relative abundance of bacterial phyla in the cecal microbiota of layers in different groups. %.

Items	Groups ^1^			SEM	*p*-Value
	CON	FA	FB		
Bacteroidetes	52.49 ^b^	58.96 ^a^	59.63 ^a^	0.01	0.04
Firmicutes	31.00	35.23	28.33	0.02	0.24
Fusobacteria	0.70	1.19	0.44	<0.01	0.48
Proteobacteria	6.02	4.22	4.91	<0.01	0.07

^a,b^ Means within each row with different superscripts are statistically significantly different (*p* < 0.05). ^1^ CON, control group; FA, basal diet with 6% FF produced by *B. subtilis*, *Lactobacillus*, *Yeast*; FB, basal diet with 6% FF produced by *C. butyricum*, *L. salivarius*.

## Data Availability

Informed consent was obtained from all subjects involved in the study. The datasets analyzed in the present study are available from the corresponding author on reasonable request.

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
