# Peer review of "Effects of Different Fermented Feeds on Production Performance, Cecal Microorganisms, and Intestinal Immunity of Laying Hens"

_animals, 2021, doi:10.3390/ani11102799_

Round 1

Reviewer 1 Report

In the present paper I reviewed, the Authors have investigated an interesting topic related to the effects of different fermented feed of laying hens. As the Authors concluded, fermented feed can decrease feed conversion ratio, improve performance and intestinal immunity of laying hens, which may be related to the improvement of cecal microflora structure.

I would like to congratulate Authors for the good-quality of their article, the literature reported used to write the paper, and for the clear and appropriate structure.

The manuscript is well written, presented and discussed, and understandable to a specialist readership.

In general, the organization and the structure of the article are satisfactory and in agreement with the journal instructions for authors.

The subject is adequate with the overall journal scope. The work shows a conscientious study in which a very exhaustive discussion of the literature available has been carried out.

The introduction provides sufficient background, and the other sections include results clearly presented and analyzed exhaustively.

However, as specific comment, with the aim to further improve the quality of the paper, the Conclusion section could be improved.

Also, the Authors have to check if alle references have been cited in the text.

So, I recommend the acceptance of the paper after minor revision.

Author Response

Dear reviewer:

We appreciate your positive comments on this article. Your encouragement is very important and meaningful to us. We have revised the article in strict accordance with the format of the magazine. Thank you again for your affirmation and encouragement to us.

Best regards

Reviewer 2 Report

In few places the language is strange, gramatically correct but consequtive sentences are very simple without relationship between them. Some spelling mistakes were also found.

Line 74-75 – was this previous research published? If so the proper quotation should be added.

The aim of study must be precised.

Line 128 – what scale was used for yolk color determination? Roche?

Line 162 – These indexes should be explained and probably their formulas should be given.

Line 172-173 – the description of statistical analysis should be combined together (with this part in lines 166-168). The proper quotation to SPSS should be added according to recommendation of IBM Corp. What kind of post-hoc test was used? It seems that particular means were compared and differences were marked by superscripts.

Table 4 – the unit of shell thickness is given mistakenly. What is the unit of yolk color? Points?

Table 5 – all abbreviations used in the table should be explained in legend, the same in case other tables.

It is very difficult to read all figures.

The list of references must be formatted according to journal’s instruction (i.e. no 3, 5, 39 etc.).

Author Response

Your comments and suggestions are greatly appreciated and detailed answers are in the attachment.

Reviewer 3 Report

The current study has evaluated the effects of different fermented feeds on production performance, cecal microorganism and intestinal immunity of laying hens. The experiment has been well designed and data has been well presented. However, the following revision could improve the quality of the paper.

Major concerns

1) English need to be improved by a native English Speaker.

  • Line 74. This sentence needs reference(s), and need to give what kind bacteria was used.
  • Line116-123. Please give the laying rate of bird at the beginning of the trial.
  • Line 87-100. It is necessary to give the collection time of feed samples and how to calculate the changes before and after fermentation, and the statistical analysis of fermented feed on a weekly basis.
  • Line 125. How the production data were collected and analyzed, such as “calculated average daily feed intake (Line 126)” and “feed conversion ratio (Line 127)”.
  • Line 135. Need more detail information for intestinal morphological analysis, such as the thickness of the sections, what was it here, tissue samples or intestinal contents?
  • In discussion. Probiotics in fermented feed were mentioned frequently. Whether the viable count after fermentation has been measured.

Minor concerns

  • Line 38. Change “has” to “have”. Please pay attention to similar questions in the passage
  • Line 66. Use of italic for “Lactobacillus”
  • Line 173. Delete “A” before “P-value of less than 0.05”
  • Table 3 was the nutrients composition of feed before or after fermentation, so the “CON" should be changed to” unfermented feed".
  • Line 176-179. Significance labeling was not standard. After check carefully, add (P< 0.05) or (P> 0.05)
  • Line 202. Add “the” before “concentration”
  • Line 225. What kind samples were used (Line130)? “Cecal tissue samples or cecal contents?”
  • Line 239. It’s better to add “From what has been discussed above,” before “compound probiotics II”
  • Line 248. Replace “ted” with “tended”
  • Line 402. Delete “FF”
  • Line 413 and 415. Modify the reference format
  • Line 416. Use of italic for “Roseburia, Butyricicoccus, Intestinimonas” and “Faecalibactium”

Round 2

Reviewer 3 Report

Thanks for your clarification. No further comment.